# Early Outcomes of Right Ventricular Pressure and Volume Overload in an Ovine Model

**DOI:** 10.3390/biology14020170

**Published:** 2025-02-07

**Authors:** Hamida Al Hussein, Hussam Al Hussein, Marius Mihai Harpa, Simina Elena Rusu Ghiragosian, Simona Gurzu, Bogdan Cordos, Carmen Sircuta, Alexandra Iulia Puscas, David Emanuel Anitei, Cynthia Lefter, Horatiu Suciu, Dan Simionescu, Klara Brinzaniuc

**Affiliations:** 1Doctoral School, George Emil Palade University of Medicine, Pharmacy, Science and Technology of Targu Mures, 38 Gheorghe Marinescu Street, 540142 Targu Mures, Romania; hamida.alhussein@yahoo.com; 2Department of Anesthesiology and Critical Care, Clinical County Hospital Mures, 1 Gheorghe Marinescu Street, 540103 Targu Mures, Romania; 3Department of Anatomy and Embryology, George Emil Palade University of Medicine, Pharmacy, Science and Technology of Targu Mures, 38 Gheorghe Marinescu Street, 540142 Targu Mures, Romania; cynthia_lefter@yahoo.com (C.L.); klara_branzaniuc@yahoo.com (K.B.); 4Regenerative Medicine Laboratory, Center for Advanced Medical and Pharmaceutical Research, George Emil Palade University of Medicine, Pharmacy, Science and Technology of Targu Mures, 38 Gheorghe Marinescu Street, 540142 Targu Mures, Romania; marius.harpa@umfst.ro (M.M.H.); exoticvet_umf@yahoo.com (B.C.); dsimion@clemson.edu (D.S.); 5Department of Cardiovascular Surgery, Emergency Institute for Cardiovascular Diseases and Transplantation, 50 Gheorghe Marinescu Street, 540136 Targu Mures, Romania; alexandra.stoica92@yahoo.com (A.I.P.); anitei_emanuel@yahoo.com (D.E.A.); horisuciu@gmail.com (H.S.); 6Department of Surgery, George Emil Palade University of Medicine, Pharmacy, Science and Technology of Targu Mures, 38 Gheorghe Marinescu Street, 540142 Targu Mures, Romania; 7Department of Pediatrics III, George Emil Palade University of Medicine, Pharmacy, Science and Technology of Targu Mures, 38 Gheorghe Marinescu Street, 540142 Targu Mures, Romania; simina_r88@yahoo.com; 8Department of Pediatric Cardiology, Emergency Institute for Cardiovascular Diseases and Transplantation, 50 Gheorghe Marinescu Street, 540136 Targu Mures, Romania; 9Department of Morphopathology, George Emil Palade University of Medicine, Pharmacy, Science and Technology of Targu Mures, 38 Gheorghe Marinescu Street, 540142 Targu Mures, Romania; simona.gurzu@umfst.ro; 10Department of Morphopathology, Emergency Clinical County Hospital, 50 Gheorghe Marinescu Street, 540136 Targu Mures, Romania; 11Experimental Station, George Emil Palade University of Medicine, Pharmacy, Science and Technology of Targu Mures, 38 Gheorghe Marinescu Street, 540142 Targu Mures, Romania; 12Department of Anesthesiology and Critical Care, Emergency Institute for Cardiovascular Diseases and Transplantation, 50 Gheorghe Marinescu Street, 540136 Targu Mures, Romania; carmensircuta@gmail.com; 13Biocompatibility and Tissue Regeneration Laboratory, Department of Bioengineering, Clemson University, Sikes Hall, Clemson, SC 29634, USA

**Keywords:** ovine model, right ventricular failure, pulmonary artery banding, pulmonary regurgitation

## Abstract

Right ventricular (RV) failure is a major complication in many types of congenital heart disease (CHD) and often leads to higher illness and mortality rates among affected patients. There are currently no specific treatments for RV failure, and therapies that are effective in left heart failure are inefficient in right heart failure due to differing adaptive mechanisms. This study aimed to develop a practical model of RV failure in sheep by creating increased pressure and volume overload, mimicking what occurs in human patients with CHD. Fourteen juvenile sheep underwent one of three surgical procedures (pulmonary artery banding, pulmonary leaflet perforation, or pulmonary annulotomy with transannular patching), each performed on a beating heart under general anesthesia with advanced anesthetic monitoring. Successful acute pressure and volume overload were obtained, replicating key features of RV failure. We developed detailed protocols for the anesthetic and surgical approach, describing the intraoperative and immediate postoperative complications. All techniques demonstrated safety and feasibility, with low mortality rates and manageable complications. Our model of RV pressure and volume overload provides a structured approach for investigating this condition, which may enhance understanding and contribute to better treatment options for patients with CHD and RV failure.

## 1. Introduction

Considering that cardiovascular disease is the leading cause of mortality worldwide, there have been tremendous advances and extensive research regarding left ventricular failure, as the left ventricle (LV) is the primarily affected ventricle in the adult population. However, the findings regarding the pathophysiology of LV failure cannot be accurately extrapolated to the pediatric population, as most cases of congenital heart disease (CHD) primarily affect the right ventricle (RV) [1,2,3].

Right ventricular failure is a common feature in multiple CHDs and increases the morbidity and mortality rates among these patients. The major determinants of non-ischemic RV failure in CHD are pressure and volume overloads on the RV [4].

A right ventricular pressure overload occurs in various situations, including pulmonary hypertension (PHT) and congenital pulmonary stenosis, or in conditions where the RV becomes the systemic ventricle (hypoplastic left heart syndrome, congenitally corrected transposition of the great arteries (ccTGA), or TGA that requires a Mustard and Senning operation) [1,5]. Alternatively, it can occur as a postoperative complication in patients with TGA who present postoperative pulmonary stenosis after an arterial switch operation (ASO) [6]. Meanwhile, creating an RV pressure overload via pulmonary artery banding (PAB) can be used as a therapeutic approach in patients with dilated cardiomyopathy [7] or as a method to train the sub-pulmonic morphologic LV in patients with TGA after the neonatal period when they require a two-staged ASO [8]. However, a right ventricular volume overload is most commonly a consequence of atrial septal defects, pulmonary regurgitation (PR), or tricuspid regurgitation (TR) [1].

PR is a common complication after the surgical correction of tetralogy of Fallot (TOF) or pulmonary valvotomy for pulmonary stenosis; more rarely, it can be seen as isolated congenital PR [6,9]. PR after TOF surgery is a growing issue among the pediatric population. Due to the advances achieved in cardiac surgery, these patients often present favorable short-term outcomes after TOF repair; however, the long-term outcomes are problematic due to the resultant PR, which becomes a major cause of morbidity and mortality [3]. Despite being well tolerated for many years, PR eventually leads to RV failure, arrhythmia, and sudden death [3,10]. Although pulmonary valve replacement (PVRpl) is an option, its timing remains a challenge, as the valve might quickly become insufficient in the growing child [3], and valved conduits have limited life duration [11]. Furthermore, it is unclear whether PVRpl fully counteracts RV remodeling, as some patients continue to present RV dysfunction after PVRpl [3,12]. Therefore, the mechanisms involved in the remodeling and reverse remodeling of the RV are not yet fully known [12].

Currently, there are no therapeutic options that specifically target the failing RV [13], and the common therapeutic options proven effective in LV failure are not usually effective in RV failure [2], as the adaptive mechanisms are different [14].

The deleterious effects of RV failure on morbidity and mortality in cardiovascular disease patients, as well as the lack of therapeutic options that directly target RV failure, have led to growing interest in understanding the pathophysiology of RV failure using animal models [13].

## 2. Aims

In this study, we aimed to develop a detailed surgical and anesthetic protocol for inducing RV failure in an ovine model by creating a pressure overload through PAB and a volume overload through pulmonary annulotomy and transannular patching (TAP), as well as pulmonary leaflet perforation.

Furthermore, we aimed to quantify and analyze RV hemodynamic and functional metrics in the preoperative and immediate postoperative phases, focusing on acute hemodynamic changes associated with pressure and volume overload. Additionally, we sought to describe and analyze intraoperative and early postoperative complications occurring within the first 7 days after surgery. These measurements aimed to assess the acute-phase changes induced by the interventions, providing insight into the feasibility and reproducibility of the RV failure induction protocol.

## 3. Materials and Methods

### 3.1. In Vitro Simulation of Surgical Procedures

To limit the surgical errors that could naturally occur at the beginning of the learning process and therefore limit the number of animals used, as well as to test the feasibility of the novel technique (pulmonary leaflet perforation using an aortic punch), we simulated the surgical procedures in vitro. Therefore, we purchased two hearts taken from three-month-old sheep from a local slaughterhouse. The hearts were weighed, and the tricuspid and pulmonary annuluses, as well as the pulmonary artery circumference, were measured.

To enable echocardiographic assessment after the procedures, we sealed the tricuspid valve using a pericardial patch (Figure 1A) and closed the pulmonary artery (PA) using a purse string (Figure 1B), thus creating a closed RV–PA loop. To ensure continuous fluid flow through the loop (simulating blood flow), an aspiration cannula was introduced into the RV through an opening between the pericardial patch and the tricuspid annulus, secured by placing a purse string suture, and a second aspiration cannula was introduced into the PA through the purse string (Figure 1B). Both cannulas were attached to 50 mL syringes, and the blood flow was simulated by injecting water into the RV, which crossed the pulmonary valve and then reached the other syringe (Figure 1C).

The three surgical procedures were then performed, as detailed in the surgical protocol section. After each surgical procedure, the RV–PA loop was filled with water, as described above, to simulate blood flow, and epicardial echocardiography was performed. Depending on the procedure performed, a pressure gradient across the banding or a pulmonary regurgitation jet was revealed, indicating successful surgical procedures.

### 3.2. Animal Selection

A total of 14 juvenile male and female sheep (Romanian Tsigai breed), aged 12–18 weeks, weighing 18–42 kg, were purchased from a local farm and randomly underwent PAB (*n* = 6), pulmonary leaflet perforation (*n* = 4), and pulmonary annulotomy + TAP (*n* = 4). Upon acquisition from the farm, the sheep were initially housed in an outdoor shelter and were randomly divided into two cages, a pressure overload cage and a volume overload cage. The animals in the volume overload cage were then randomly divided into a pulmonary leaflet perforation cage and an annulotomy + TAP cage. Before surgery, the animals were relocated to an indoor facility, which included three individual recovery boxes used for both preoperative preparation and immediate postoperative care. There was also a postoperative containment cage, in which the animals were housed until the removal of the drainage tube. The indoor shelter was equipped with both natural and artificial lighting, as well as a heating and ventilation system to maintain optimal conditions. Air circulation, dust levels, and gas concentrations were regulated to remain within safe limits for the animals. The temperature was maintained at approximately 25 °C and relative humidity at around 60%, ensuring a controlled environment conducive to postoperative recovery. Post-surgery, a minimum of two sheep were consistently housed together indoors to accommodate the natural herd behavior of the species. After 7–10 days of postoperative care, the sheep were transferred to the outdoor cages designated for each group, where they were housed with other surgically treated sheep.

This experimental research was conducted in adherence to the 3R principles (replacement, reduction, refinement) to the greatest extent possible to ensure the animals’ welfare by reducing the number of animals to the minimum (reduction) and by ensuring proper analgesia, both intraoperatively and in the postoperative period (refinement).

All the procedures and the perioperative care of the experimental animals were performed following the “Guide for the Care and Use of Laboratory Animals” and Directive 2010/63/EU of the European Parliament on the protection of animals used for scientific purposes.

Ethical approval for this study was obtained from the Scientific Research Ethics Committee of the George Emil Palade University of Medicine, Pharmacy, Science, and Technology of Targu Mures, with approval no. 1451/22.07.2021. This research was approved by the Mures Sanitary Veterinary and Food Safety Department, with approval no. 54/31.08.2022.

### 3.3. Preoperative Preparation

Twenty-four hours before the surgery, each animal was transferred to an indoor shelter (Figure 2A). The skin from the surgical site, as well as from the venous and arterial lines area, was shaved and disinfected. The animal was weighed and clinically examined via lung and heart auscultation. Solid food was restricted 18–24 h before surgery and liquid intake was stopped 8–12 h before surgery.

The surgical and anesthetic equipment and devices were verified, ensuring their functionality (Figure 2B). In animals weighing < 30 kg, a pediatric anesthesia circuit with a 1 L rebreathing bag was used, while in animals weighing > 30 kg, an adult anesthesia circuit with a 2 L rebreathing bag was employed. Two intravenous infusion sets and two pressure transducer monitoring kits were primed with normal saline and ready for use. All anesthetic and emergency medications were prepared in appropriate concentrations, as detailed in Appendix A, and labeled in syringes. Inotropic and vasoactive agents, including dopamine, dobutamine, adrenaline, noradrenaline, and milrinone, as well as temporary epicardial pacing wires and internal defibrillation paddles, were available, if needed. Infusion solutions, including Ringer’s solution and 0.9% sodium chloride, were prepared, along with two infusion-warming devices.

### 3.4. Anesthetic Protocol

#### 3.4.1. Pre-Anesthetic Sedation

Intramuscular medetomidine and subcutaneous atropine were administered 30–40 min before the surgery to facilitate the animal’s transportation to the operating room. Atropine was added to combat the bradycardia caused by medetomidine. The medications used at different stages of anesthesia, the doses, and the routes of administration are presented in Appendix A. The sheep was then positioned on the operating table in right lateral recumbency and was secured with straps. Pre-induction anesthetic monitoring consisted of a three-lead electrocardiogram (ECG), pulse oximetry using an auricular pulse oximeter, and non-invasive blood pressure (BP) monitoring using a BP cuff placed on the right forelimb.

Peripheral venous access was obtained by placing an 18 G peripheral venous catheter into the cephalic vein of the left forelimb (Figure 2C).

Preoperative echocardiography was performed before inducing general anesthesia to determine the baseline values (Figure 2D). These included the aortic and pulmonary velocities, valvular size and function, mitral annular plane systolic excursion (MAPSE), tricuspid annular plane systolic excursion (TAPSE), left ventricular systolic function by determining the ejection fraction using the Teichholz formula, and the diastolic function of both ventricles through the evaluation of trans-mitral and trans-tricuspid flow velocities and the E/A ratios.

#### 3.4.2. Anesthesia Induction

After adequate preoxygenation with 100% O_2_ through a facial mask at a flow rate of 8 L/min for 3–5 min, the following induction sequence was implemented intravenously: atropine, midazolam, ketamine, propofol, and atracurium. The doses used are detailed in Appendix A. Following adequate neuromuscular blockade, a laryngoscopy was performed using a size 5 Miller blade with a 4 cm extension. Endotracheal (ET) intubation was performed using a cuffed ETT, loaded on a stylet and introduced through the mouthpiece (Figure 3A). The ETT’s internal diameter varied from 6 to 7.5 mm, depending on the size and weight of the animal. Correct ET intubation was verified through bilateral lung auscultation and the presence of a capnography waveform. After intubation, the pulse oximeter was switched to a lingual pulse oximeter for a more accurate reading (Figure 3A).

Intraoperative prophylactic antibiotic therapy consisted of ceftriaxone, a third-generation cephalosporin. A test dose consisting of 10% of the total dose was administered during induction, while the remainder was administered after one hour, if no allergic reaction was noted. The dose used is included in Appendix A.

Mechanical ventilation was initiated using the pressure-controlled ventilation mode with a tidal volume (TV) of 7–8 mL/kg and a respiratory rate of 12–20 breaths/minute to maintain end-tidal CO_2_ (EtCO_2_) within the range of 30–40 mmHg (Figure 3B).

A nasogastric tube was used only if ruminal tympany occurred. We found that a classic nasogastric tube could easily become tangled or clogged with the remaining ruminal content, despite its size. A Faucher tube was generally avoided for safety reasons, especially in smaller animals. Moreover, we found that if the gastric content was aggressively aspirated during the surgery, in the following postoperative days, the animal presented meteorism, low feeding tolerance, and constipation. Regurgitation and aspiration pneumonia upon intubation can be successfully avoided if the nothing by mouth (NPO) recommendations are respected. Therefore, in our study, if an animal presented ruminal tympany during the surgery and a nasogastric tube could not be placed successfully, a ruminal puncture using a 16 G peripheral venous catheter was performed at the end of the surgery, with successful decompression.

The post-induction anesthetic monitoring included the addition of EtCO_2_ monitoring; monitoring of the central body temperature, invasive BP, and central venous pressure (CVP); invasive hemodynamic monitoring using the Swan–Ganz catheter; and urine output monitoring.

Central body temperature monitoring was achieved by inserting an esophageal temperature probe through the nasal cavity to the upper esophagus (Figure 3A). To maintain normothermia during the surgery, a heated operating table was used, the infusion solutions were warmed, and the operating room’s temperature was kept at 20–22 °C.

Arterial lines were used for invasive BP monitoring during the surgery and were left in place after the surgery for 4 to 7 days for daily arterial blood gas analyses. Due to the low durability of arterial catheters in the postoperative period, we placed two arterial lines in each animal, with one of them serving as a backup. Under sterile conditions, one arterial line was placed in the left auricular artery using a peripheral venous catheter with a diameter of 20 G and a length of 3 cm (Figure 3C,D). The other arterial catheter, with a diameter of 20 G and a length of 8 cm, was placed in the left tibial artery (Figure 3E,F). In the postoperative period, both lines were flushed daily with heparinized 0.9% sodium chloride and locked with 0.8 mL of heparin each to prolong their patency, and the dressing was changed every other day.

A central venous catheter (CVC) was placed under sterile conditions, using a three- or four-lumen CVC with a diameter of 7–8.5 Fr and a length of 20 cm, inserted into the left external jugular vein (Figure 4A,B). During the surgery, the distal lumen was used for continuous central venous oxygen saturation (ScvO_2_) monitoring. After the surgery, the catheter was left in place for 7–10 days for postoperative drug administration and periodic venous blood gas and serum electrolyte monitoring, as well as other analyses, if needed.

Central venous oxygen saturation (ScvO_2_) was monitored continuously using the CeVOX optical module (Mindray^®^, Shenzhen, China) by placing a fiber optic probe through the distal lumen of the CVC (Figure 4A,B).

The Swan–Ganz catheter was placed under sterile conditions using an 8.5 Fr introducer sheath inserted into the left external jugular vein, 1–2 cm above the CVC insertion site, through which a 7.5 Fr Swan–Ganz catheter with 5 lumens was inserted (Figure 4A,B). The catheter was connected to the cardiac output (CO) module of the monitor. The distal lumen was connected to a transducer for continuous RV and PA pressure monitoring. Additionally, the proximal lumen was connected to a transducer for CVP monitoring. The in-line injection temperature sensor was connected to the injectate port and the temperature probe. After placement, the catheter was advanced into the PA (Figure 4C), with the balloon inflated until a wedge pressure waveform was obtained. The right atrial, right ventricular, pulmonary artery, and pulmonary artery wedge pressures (PAWP) were registered, along with the concomitant systemic BP, heart rate, and peripheral oxygen saturation (SpO_2_). The injection syringe was then attached for the thermodilution measurement of CO (Figure 4D), which was performed using 3 × 10 mL of cold 0.9% sodium chloride. The distance from the insertion site to the PA was approximately 40–55 cm, depending on the animal’s size. All mentioned parameters were registered before and after the surgical procedure was completed.

The urine output was easily monitored without the need for a urethral catheter due to the specialized two-panel V-top veterinary operating table used, which was equipped with a waste channel and drain hole. To monitor the urine output during the surgery, a graded vessel was placed underneath the drain hole.

#### 3.4.3. Anesthesia Maintenance

Throughout the surgery, anesthesia was maintained using inhaled sevoflurane at a concentration of 1.5–2.5%, with a fresh gas flow rate of 2.5 L/min (0.9 L/min oxygen in 1.6 L/min air) and a fraction of inspired oxygen (FiO_2_) of 50%. Intermittent intravenous ketamine and atracurium were administered at 30–40 min intervals, with the doses detailed in Appendix A. We used ketamine during the induction and maintenance of anesthesia due to its analgesic effects; it was used as a substitute for fentanyl, as opioids were difficult to purchase due to legal restrictions.

Advanced anesthetic monitoring was ensured during the surgery (Figure 4E,F). Arterial and venous blood gas analyses, as well as a serum electrolyte panel, were performed before and after the procedure, and imbalances were corrected as needed.

#### 3.4.4. Animal Awakening from Anesthesia

Towards the end of the surgery, to counteract the progressive atelectasis produced by the rib spreader’s application and prolonged lateral recumbency, after the removal of the rib spreader, a manual alveolar recruitment maneuver was performed by switching the ventilator to bag mode, setting the adjustable pressure-limiting (APL) valve to 35–40 cmH_2_O, and delivering a few breaths. Additionally, sterile endotracheal aspiration was performed, followed by oropharyngeal aspiration to ensure a clear airway.

Before anesthesia reversal, at the end of the surgery, paracetamol and ketorolac were administered to ensure postoperative analgesia, along with metoclopramide to combat postoperative nausea, dexamethasone to decrease postoperative inflammation, and furosemide, depending on the urine output and intraoperative fluid balance. The doses used are detailed in Appendix A.

The Swan–Ganz catheter and introducer sheath, the ScvO_2_ fiber optic probe, and the esophageal temperature probe were removed. BP monitoring was switched to non-invasive monitoring, and the pulse oximeter was switched to an auricular pulse oximeter.

The last dose of atracurium was administered 30–40 min before awakening. After skin closure, the administration of sevoflurane was stopped, and the animal was ventilated with 100% O_2_ at a flow rate of 8–8.5 L/min. Mechanical ventilation was switched to manual ventilation to increase the EtCO_2_ and trigger spontaneous breathing. Awakening was evaluated in terms of the presence of the ciliary reflex, eye opening, swallowing, and spontaneous breathing. If awakening was noted, the animal was extubated after meeting the following criteria: (1) adequate spontaneous breathing, with a TV ≥ 6 mL/kg and a respiratory rate of ≥12/min; (2) adequate oxygenation, with an SpO_2_ of ≥93% at an FiO_2_ of ≤40%; (3) adequate neuromuscular blockade recovery (sustained head lift for 5 s); (4) hemodynamic stability.

After extubation, oxygen was provided through a facial mask as needed, with a flow rate of 8.5 L/min, and the animal was further monitored for 5–10 min. If the SpO_2_ level was ≥93% in ambient air and the animal was fully awake and hemodynamically stable, the peripheral venous catheter was removed, followed by slight hemostatic compression, and the remaining anesthetic monitoring was stopped. The animal was then transported to a postoperative containment cage, placed in ventral recumbency, and continuously monitored until it achieved sustained, balanced orthostatism. In the following hours, the state of consciousness, presence of rumination, urine and fecal production, respiratory pattern, drainage tube patency, and drained fluid quantity and aspect were monitored. Water was offered approximately two hours after awakening, while food was progressively reintroduced six hours after the surgery.

#### 3.4.5. Surgical Protocol

A 10 cm skin incision was created in the third/fourth left intercostal space (based on a preoperative evaluation of thoracic anatomy and the animal’s weight), followed by a left thoracotomy. A rib spreader was placed, and the left internal mammary artery and vein were identified and avoided. Blunt dissection was performed to separate the intercostal muscles, minimizing tissue damage. Hemostasis was maintained using bipolar cautery. The lower left pulmonary lobe was moved laterally for better access. To expose the right heart, a “reverse T” pericardiotomy was performed, avoiding the phrenic nerve (Figure 5A). The pericardium was retracted and secured to the thoracic wall with 2.0 stay sutures to maintain a stable and unobstructed surgical field. As all procedures were performed off-pump, systemic heparin was not needed.

##### Pulmonary Artery Banding (PAB)

After exposing the PA (Figure 5A) and measuring its circumference, a Gore-Tex band was prepared (Figure 5B). The band was pre-soaked in normal saline and trimmed to a width of 5 mm to ensure flexibility and ease of placement. The initial length of the band was determined using Trusler’s formula (20 mm + 1 mm/kg) [15], with subsequent adjustment as needed. A surgical caliper was used to confirm the dimensions of the band and its compatibility with the measured PA diameter. The diameter of the PA was narrowed by progressively placing 2.0 Ti-Cron sutures around the Gore-Tex band (Figure 5C,D), under continuous monitoring of invasive BP and RV pressure through the Swan–Ganz catheter. We targeted an RV pressure of approximately 2/3 of the systemic BP, depending on the hemodynamic tolerance of the animal. If the animal became hemodynamically unstable, the last suture placed was loosened, the RV was given more time to adapt, and the suture was tightened again. If the hemodynamic instability was sustained and refractory, the suture was removed, and another one was placed a few millimeters distally. This process was repeated until the highest RV pressure with the lowest hemodynamic instability was achieved. The band was positioned approximately 2 cm proximal to the pulmonary bifurcation to ensure uniform pressure distribution. To ensure stability and prevent migration, the band was anchored to the adventitia of the pulmonary artery, proximally and distally, using fine 4.0 polypropylene stitches.

##### Pulmonary Leaflet Perforation

During systole, the PA and the right ventricular outflow tract (RVOT) were partially clamped using a Satinsky vascular clamp (Figure 6A). The clamp was closed during systole to ensure that the pulmonary valve was open, with the leaflets apposed to the arterial wall. This allowed for precise manipulation and access to the anterior pulmonary leaflet. The clamp was carefully positioned with a minimum of 1 cm proximal to the pulmonary valve annulus to provide optimal exposure, without excessive compression of the surrounding structures. After achieving hemodynamic stability, a longitudinal incision of approximately 3 cm was created at the level of the PA root, exposing the anterior pulmonary leaflet (Figure 6A). Using a 6 mm diameter aortic punch, the pulmonary leaflet was perforated (Figure 6B). Care was taken to avoid damaging adjacent leaflet tissue or the pulmonary artery wall to ensure that the procedure was confined to the target leaflet. After perforation, the edges of the punched leaflet were inspected and cleared of any clots or tissue remnants to prevent flow obstruction. The PA was then sutured using a double-layered continuous suture with a 4.0 Prolene thread. The sutures were placed and tied carefully to avoid altering the diameter of the pulmonary artery, ensuring that no stenosis or obstruction to blood flow occurred as a result of the arteriorrhaphy. After de-airing the PA through a small puncture at the distal suture line, the Satinsky clamp was removed gradually to prevent abrupt hemodynamic fluctuations.

##### Pulmonary Annulotomy and Transannular Patching (TAP)

During diastole, the PA and RVOT were partially clamped using a Satinsky vascular clamp, as described above. The clamp was closed during diastole to ensure that the pulmonary valve was fully closed, providing optimal access to the valvular annulus for annulotomy. This positioning allowed the annulotomy to be performed precisely along the annulus and at the base of the anterior leaflet. The clamp position was monitored to ensure minimal compression while maintaining adequate exposure for the incision. Under hemodynamic stability, a longitudinal incision of approximately 3–5 cm was created at the level of the PA, pulmonary annulus, and RVOT (Figure 6C). The incision was carefully extended along the RVOT, specifically through the infundibular wall proximal to the pulmonary valve, while avoiding damage to the right coronary artery or further extension onto the free wall of the right ventricle to prevent structural weakening, bleeding, or complications with TAP placement. Transannular patching was performed using a heterologous pericardial patch with a length of 3–5 cm and a width of approximately 2 cm, which was sutured using a 5.0 Prolene continuous suture (Figure 6D). The patch was pre-treated with glutaraldehyde (GA) for 10 min to enhance its durability and resistance to calcification. The 10 min treatment duration was chosen based on its practicality in the surgical setting and evidence suggesting that shorter fixation times can achieve sufficient crosslinking while preserving tissue flexibility. Lee et al. demonstrated that while increasing the GA concentration significantly reduced calcification, the fixation time itself did not have a significant effect on calcification resistance [16].

The sutures were placed at 1 mm intervals to ensure a secure and watertight closure. After completing the suture line, the patch was inspected for potential leaks, and any identified were immediately reinforced with additional sutures. After proper de-airing, the Satinsky clamp was removed.

At the end of the procedure, arterial blood gas analysis and a serum electrolyte panel were performed, and a new set of invasive measurements was recorded using the Swan–Ganz catheter (Figure 7A).

After achieving hemostasis and reviewing the cardiotomy sites, a left pleuropericardial drain tube was placed through a separate incision. After removing the rib spreader, an alveolar recruitment maneuver was performed, as mentioned above. Partial pericardial closure was performed, followed by rib adduction with isolated sutures, closure of the surgical incision in anatomical layers, and local application of a silver-containing skin antiseptic.

#### 3.4.6. RV Hemodynamic and Functional Metrics

Demographic data for all animals, including age (weeks), weight (kg), height (cm), and body surface area (BSA) (m^2^), were recorded preoperatively.

To evaluate the immediate effects of the surgical procedures on RV function, preoperative and immediate postoperative RV metrics were measured and analyzed. Hemodynamic parameters, measured using a Swan–Ganz catheter placed via the left external jugular vein, included the following: (1) RV pressures—systolic (sRVP), diastolic (dRVP), and mean (mRVP); (2) pulmonary artery pressures—systolic (sPAP), diastolic (dPAP), and mean (mPAP); (3) central venous pressure (CVP) and pulmonary artery wedge pressure (PAWP); (4) cardiac performance indices—cardiac index (CI) and stroke volume index (SVI); (5) systemic and pulmonary vascular resistances—systemic vascular resistance index (SVRI) and pulmonary vascular resistance index (PVRI).

Additionally, arterial and venous blood gas analyses were performed pre- and immediately post-surgery to monitor oxygenation, acid–base balance, and lactate and electrolytes levels, as indicators of RV adaptation and systemic effects. These included central venous oxygen saturation (ScvO_2_); partial pressures of oxygen (pO_2_) and carbon dioxide (pCO_2_); and bicarbonate (HCO_3_^−^), base excess (BE), and serum lactate levels.

#### 3.4.7. Intraoperative Epicardial Echocardiography

After the surgical procedure had been completed, epicardial echocardiography was performed using a sterile covered cardiac transducer (Figure 7B) and a Mindray^®^ ultrasound machine. The global contractility was assessed, and depending on the procedure performed, the maximum and mean gradients across the PAB and the degree of pulmonary regurgitation were evaluated (Figure 7C).

#### 3.4.8. Postoperative Care

During the first seven postoperative days, a standardized monitoring protocol was implemented, consisting of a detailed clinical examination, as presented in Appendix A, which was performed daily. Moreover, arterial and venous blood gas analyses, as well as serum electrolyte panels, were performed daily to guide the therapy. The chest drain tube was monitored daily and removed 1–4 days post-surgery, depending on the amount of fluid drained. The central venous and arterial catheters were maintained as described above. During the first postoperative week, the treatment consisted of antibiotics, loop diuretics, analgesics, antipyretics, and steroids in all cases, as well as other symptomatic and adjuvant treatments that were individualized for each case, as needed. The doses used and the routes of administration, as well as the administration frequencies, are described in Appendix A.

#### 3.4.9. Postoperative Transthoracic Echocardiography

During the immediate postoperative period, transthoracic echocardiography was performed on the third and seventh postoperative days. In the absence of complications, such as the presence of significant pericardial effusion, the CVC was removed, and the animal was moved to an outdoor shelter with other sheep that had undergone operations.

After moving the animal to the outdoor facility, serial echocardiography was performed monthly to assess the systolic and diastolic cardiac function and monitor the onset and progression of RV failure.

#### 3.4.10. Statistical Analysis

All data are presented as mean ± standard deviation and medians. Statistical analysis was carried out using GraphPad Prism software, version 10.4.0 (GraphPad Software, 225 Franklin Street. Fl. 26, Boston, MA, USA). Since the group sizes were too small to reliably run tests on Gaussian distribution, nonparametric methods were chosen. The Wilcoxon signed-rank test was used for comparison between the pre- and post-surgical hemodynamic parameters and blood gas analyses within each group. For comparison of demographic data (age, weight, height, and body surface area) across all groups, the Kruskal–Wallis test with Dunn’s multiple comparison post hoc test was applied. Statistical significance was assumed for *p* < 0.05.

## 4. Results

We were able to create standardized acute pressure overload via PAB and acute volume overload via annulotomy with TAP augmentation, as well as pulmonary leaflet perforation, in all animals.

A mean value of the peak pressure gradient of 44.3 mmHg, evaluated by echocardiography, was achieved in the PAB group. Additionally, moderate-to-severe pulmonary regurgitation was achieved in both the TAP and leaflet perforation groups. The grade of pulmonary regurgitation was quantified using echocardiographic measurement of the PR jet width/RVOT diameter ratio, which was 0.58% in the TAP group and 0.51% in the leaflet perforation group.

### 4.1. RV Hemodynamic and Functional Metrics

There were no significant differences between the groups regarding age, weight, height, or body surface area (BSA) (see Appendix A).

The hemodynamic effects of acute PAB, TAP, and pulmonary leaflet perforation are detailed in Appendix A.

Acute pressure overload resulted in a 2.2-fold (120%) increase in systolic RV pressure (*p* = 0.0312), a 3-fold increase in diastolic RV pressure (*p* = 0.0312), and a 2.5-fold increase in mean RV pressure (*p* = 0.0312). Moreover, a significant decline in the systolic and mean PA pressures (*p* = 0.0312), as well as the pulmonary vascular resistance index (PVRI) (*p* = 0.0312), was noted. Additionally, PAB led to a significant 40% reduction in the stroke volume index (SVI) (*p* = 0.0312) and a 42% in the cardiac index (CI) (*p* = 0.0312), alongside a significant decline in both the left ventricular stroke work index (LVSWI) (*p* = 0.0312) and the right ventricular stroke work index (RVSWI) (*p* = 0.0312).

Acute volume overload, on the other hand, resulted in a slight increase in the systolic, diastolic, and mean RV pressures, without reaching statistical significance.

An incremental trend in SVI and CI was observed across both groups. Specifically, SVI increased by 24% in both the TAP and leaflet perforation groups, while CI increased by 23% in the TAP group and 13% in the pulmonary leaflet perforation group. However, these changes did not reach statistical significance, likely due to the small sample size.

The effects of acute PAB, TAP, and pulmonary leaflet perforation on the ScvO_2_, arterial blood gas analysis, and serum electrolyte panel are depicted in Appendix A.

Acute pressure overload resulted in a significant decline in ScvO_2_ (*p* = 0.0312) and a significant elevation in serum lactate levels (*p* = 0.0312). In contrast, these parameters did not differ significantly in the volume overload groups.

Statistical analyses for the hemodynamic and metabolic measurements registered before (pre-op) and immediately after (post-op) PAB were performed using the Wilcoxon signed-rank test for non-parametric data, with a *p*-value of <0.05 considered significant. The results of these analyses for the parameters that demonstrated significant differences are graphically represented in Figure 8, Figure 9 and Figure 10.

### 4.2. Postoperative Care

Antibiotic therapy was continued with ceftriaxone on the first postoperative day, followed by ceftiofur, a third-generation cephalosporin for veterinary use, administered until the CVC was removed. Prophylactic metronidazole and fluconazole were administered in one case in which the chest tube was accidentally disconnected by the animal.

Loop diuretics (furosemide) were administered, depending on the clinical status, echocardiographic aspect, serum potassium levels, and presence of pericardial/pleural effusions.

The analgesic treatment included tramadol, administered twice a day until the chest tube was removed, after which it was administered once a day, as well as metamizole. The antipyretics included paracetamol, which was administered in cases of fever.

Steroid anti-inflammatory treatment with dexamethasone was administered at a full dose on the first three postoperative days, followed by a gradual dose reduction over two weeks. Steroid tapering consisted of reducing the dose by 50% every three days. After CVC removal, dexamethasone was given orally to complete the tapering process. In cases of significant pericardial effusion after chest tube removal, the treatment with dexamethasone was prolonged.

Depending on the animal’s clinical evolution and daily blood analyses, treatment with spironolactone and/or oral potassium was administered, based on the serum potassium levels. In case of bloating, simethicone was administered orally. Moreover, acetazolamide was given orally to manage metabolic alkalosis, while ibuprofen was administered in case of pericardial effusion after removing the chest drain.

Depending on the clinical evolution of each animal, the treatment was prolonged, if needed. The doses used and the routes of administration, as well as the administration frequencies, are described in Appendix A.

### 4.3. Complications

The intraoperative and early postoperative complications are detailed in Table 1 and Table 2, while the chest drainage period and total drained quantities are provided in Appendix A.

Among the observed intraoperative complications, arrhythmias were the most frequent, especially during the procedure (Table 1). Most of these did not have a hemodynamic impact and did not require pharmacological treatment.

Despite pre-medication with subcutaneous atropine, sinus bradycardia was observed in 10 out of 14 animals in the pre-induction period; therefore, after obtaining venous access, intravenous atropine was administered as needed.

During the procedure, most arrhythmias were observed in the PAB group (Table 1). Bradycardia associated with ST segment elevation and hemodynamic instability was observed in four out of six animals during band tightening and was resolved only by loosening the band.

Atrial fibrillation (AF) was observed in one PAB animal, which was self-limiting and converted to a sinus rhythm spontaneously, respectively, in one annulotomy + TAP animal, which required amiodarone for conversion to sinus rhythm (5 mg/kg administered over 20 min).

Hypothermia was observed in 12 out of 14 animals, despite using warm intravenous solutions and a heated operating table. The core body temperature tended to drop after chest opening and increased after chest closure. Hyperthermia (41.5 °C) was observed in one animal and was resolved by switching off the heated operating table.

Tympanism was observed in six cases, despite the strict NPO regimen. It was likely caused by the prolonged lateral recumbency and favored by the medetomidine administration, which decreases gastrointestinal motility. It was resolved by either placing a nasogastric tube or by performing a ruminal puncture at the end of the procedure.

Regarding the early postoperative complications (during the first week) (Table 2), we registered one early death, which occurred within less than 12 h postoperatively, in one PAB animal. During the procedure, the animal presented AF, which converted spontaneously to a sinus rhythm. The necropsy and histopathology exam yielded nonsignificant results; therefore, we suspected an arrhythmic event as the cause of death.

Significant pericardial effusion that resulted in cardiac tamponade was registered in one animal that underwent annulotomy + TAP. During echocardiography, the animal presented cardiac arrest and was successfully resuscitated after 5 min of cardiopulmonary resuscitation (CPR). The effusion was ultimately resolved by prolonging the treatment with oral dexamethasone, followed by switching to oral ibuprofen, as well as aggressive diuretic treatment with furosemide and spironolactone.

A pneumothorax was registered in one animal that accidentally disconnected its chest tube on the first postoperative day. The chest tube was disinfected and reconnected, and the animal received empiric antibiotic treatment with ceftiofur, metronidazole, and fluconazole for seven days, with a favorable outcome.

The other postoperative complications were mild and were easily resolved through oral treatment, as mentioned above. Self-limiting mild fever (40–41 °C) was present more frequently in the PAB group, especially on the first two postoperative days, and was likely caused by inflammation, along with a possible component of low cardiac output.

The annulotomy + TAP group required a longer period of chest drainage, and the drained quantity was larger than that in the other two groups (see Appendix A); this was likely due to the procedure itself, which was more invasive and required the incision of the RVOT.

## 5. Discussion

### 5.1. Pathophysiology of RV Failure

While the RV is more tolerant of changes in preload, acute changes in afterload are usually poorly tolerated [1]. An acute and significant increase in afterload can lead to a significant decrease in cardiac output (CO) and acute RV failure. This is likely due to the failure of the Anrep mechanism in increasing the RV contractility in acute conditions, which is the first-line RV adaptive mechanism to increased afterload. Moreover, the acute increase in RV filling pressures can lead to decreased diastolic blood flow in the right coronary artery, leading to RV ischemia [17].

In contrast to pressure overload, volume overload is usually tolerated by the RV for a long period of time, and it usually results in RV enlargement and RV diastolic dysfunction, characterized by increased RV end-diastolic pressure, which may persist for decades before RV systolic dysfunction occurs [3].

In acute conditions, the RV adapts to volume overload through the Frank–Starling mechanism by increasing its cavity and end-diastolic volume to increase its stroke volume (SV) and CO. However, this increase in cardiac output declines over time in chronically volume-overloaded RVs [18].

Our PAB technique for inducing RV failure through pressure overload resulted in an acute increase in systolic RV pressure by 120% (24.5 ± 6.5 mmHg vs. 53.5 ± 10.7 mmHg) (Figure 8A), which is twice the value reported by Yerebakan et al., who achieved an increase of 60% (26.8 ± 3.6 mmHg vs. 42.8 ± 4.3 mmHg) and that of Hon et al. (25.3 ± 4.0 mmHg vs. 41.4 ± 6.0 mmHg) [6,8].

The significant decrease in SVI and CI observed in our study (Figure 9C,D) is consistent with the findings of Yerebakan et al. This group reported an acute increase in end-diastolic volume (EDV) and a greater increase in end-systolic volume (ESV), leading to a significant decline in SV by 12.5% [6].

On the other hand, Hon et al. reported a significant elevation in RV output by approximately 35% after PAB. This was due to a significant increase in EDV, without a significant change in ESV [8].

The contrasting results might be explained by the tighter band obtained in our study, reaching a higher peak systolic RV pressure (53.5 mmHg vs. 41.4 mmHg in Hon’s study). This might also explain the greater reduction in SVI observed in our study compared to that in Yerebakan’s study (40% vs. 12.5%).

When faced with acute increase in afterload, the RV can increase its contractility by homeometric autoregulation to adapt to a double of its afterload, in theory, after which ventriculo–arterial (VA) decoupling occurs [19]. VA coupling refers to the interaction between the myocardial contractility, represented by the ventricular end-systolic elastance (Ees) and its afterload, represented by the effective arterial elastance (Ea) [20].

The RV operates at maximum efficiency and submaximal stroke work under control conditions, where Ees is usually greater than Ea. When faced with afterload increase and when Ea equalizes Ees, VA coupling is optimized. However, when Ea exceeds Ees, VA decoupling occurs, leading to decreased SV and CO, resulting in RV failure [21].

In line with the decline in SVI and CI in our study, we also noted a significant decrease in RVSWI, which further highlights RV impairment. Right ventricular stroke work has been shown to correlate with TAPSE, abnormal WHO class, and the need for septostomy, as well as mortality in children with PHT [22]. Additionally, our research demonstrated a notable reduction in LVSWI within the PAB group (Figure 9E), emphasizing the concept of ventricular interdependence. In line with this, Leeuwenburgh et al. observed diminished LV volumes and stroke work under chronic RV pressure overload. This is primarily attributed to a reduced LV filling caused by reduced RV output, which is likely the mechanism involved in acute settings. In chronic conditions, the shift of the interventricular septum may further decrease LV filling [23].

Moreover, we noted a significant reduction of ScvO_2_ in the PAB group (Figure 10A), as well as a significant increase in serum lactate levels (Figure 10B). This clearly reflects and highlights the reduced SVI and CI in this group. In a case report of chronic RV pressure overload, Ukita et al. documented a dramatic drop in ScvO_2_ to 42% by week 7 and 27% by week 8 postoperatively, suggesting RV decompensation, which ultimately led to animal death [24]. In a different study, the same authors suggested that a rapid banding strategy led to significant decrease in ScvO_2_ below 65%, suggesting decompensated RV failure, while a more gradual and progressive banding approach maintained the ScvO_2_ within its physiological limits of 70–80% [25].

The procedures performed in our study for inducing RV failure through volume overload resulted in moderate-to-severe pulmonary regurgitation in both the TAP and leaflet perforation groups. The severity of pulmonary regurgitation was assessed by echocardiography performed in a blinded manner by a pediatric cardiologist. It was quantified using the PR color jet width/RVOT diameter ratio, which was 0.58% in the TAP group and 0.51% in the leaflet perforation group. The echocardiographic determination of PR severity has been validated to a lesser extent than other valvular regurgitations. According to the 2010 European Association of Echocardiography (EAE) recommendations, a PR jet width that occupies > 65% RVOT diameter is classified as severe PR [26]. However, according to the 2013 European Association of Cardiovascular Imaging (EACVI) recommendations, a PR jet width that occupies > 50–65% of the RVOT diameter suggests severe PR [27]. Therefore, we classified the PR in our study as moderate-to-severe.

Similar PR severity was obtained by Yerebakan et al., who reported a moderate-to-severe (grade II–III) PR; however, the authors did not specify the quantification method used. In this study, the authors reported increased RV pressures and volumes and a significant increase in SV, by approximately 24%, and CO, by approximately 30%, in the acute phase [6].

These findings are in line with our results, as we noted an incremental trend in RV pressures, as well as an increase in SVI by approximately 24% in both the TAP group (47.3 ± 9.8 mL/m^2^ vs. 58.6 ± 15.5 mL/m^2^) and the leaflet perforation group (51.2 ± 19.1 mL/m^2^ vs. 63.2 ± 10.4 mL/m^2^). Moreover, an increase in CI of 23% in the TAP group (4.8 ± 1.7 L/min/m^2^ vs. 5.9 ± 1.6 L/min/m^2^) and of 13% in the leaflet perforation group (5.6 ± 2.3 L/min/m^2^ vs. 6.3 ± 1.2 L/min/m^2^) was observed. However, our results did not reach statistical significance, most likely due to the small animal sample size. The lower increment in CI in the leaflet perforation group, compared to that of the TAP group, might be attributed to the higher preoperative HR in the leaflet perforation group (108.3 ± 16.2 beats/min vs. 99.3 ± 19.4 beats/min in the TAP group).

### 5.2. Animal Models

The gap in knowledge regarding RV failure and the growing population of CHD patients have led to an urgent need for preclinical research. Large-animal models are especially encouraged, as they can be rapidly translated into clinical practice [28]. Such animals present the advantage of a “human-like” anatomy that facilitates surgical procedures [13]. Moreover, among large animals, sheep have become a widely used model, especially in cardiovascular research, due to their modest body mass increase and docile behavior [29], as well as their anatomical and hemodynamic similarities to human physiology [30,31].

While multiple ovine models of ischemic and non-ischemic left heart failure have been described, ovine models of right heart failure are relatively scarce [29], especially those of CHD-related RV failure.

### 5.3. Inducing RV Failure

As mentioned above, pressure and volume overload are the main underlying pathophysiological mechanisms that lead to RV failure in most CHD cases. Considering that some of the anatomical anomalies of certain CHDs cannot always be recreated in animal research settings due to their complexity, an animal model for CHD-related RV failure can be created by causing pressure and volume overload.

PAB is a commonly used technique for inducing RV pressure overload and has been described in small- and large-animal models. Its main advantage is its simple surgical technique, which does not require cardiopulmonary bypass (CPB), as well as the possibility of creating a controlled stenosis, according to the animal’s hemodynamic tolerance. On the other hand, the main challenge is achieving a PAB that is tight enough to induce RV failure and not merely compensated hypertrophy while, at the same time, being loose enough to not cause acute RV failure and death. This challenge has been overcome with the introduction of adjustable PAB [32].

Ukita et al. documented a progressive pressure overload approach by using an adjustable PAB. This method provides a significant advantage by enabling progressive tightening over time via a subcutaneous inflation port. This enables the achievement of systemic pressures in the RV, which cannot be achieved in the acute settings due to subsequent acute RV failure. This gradual increase in afterload allows for controlled RV adaptation, mimicking chronic and progressive pressure overload scenarios in humans, as seen in PHT patients [25,33].

While our approach achieved immediate and reproducible pressure gradients, it lacked the incremental adjustment capability provided by adjustable PAB. However, the simplicity of our technique reduces procedural complexity and eliminates the need for long-term management of an inflation system, which may introduce complications such as infection or malfunction.

Alternatively, a fixed PAB can be used in young animals so that the stenosis will become more severe over time as the animal grows [13].

In contrast to the multiple pressure overload animal models, there is a recognized gap in knowledge and a lack of experimental research regarding the volume-overloaded RV [13,18].

The described RV volume overload models include aortocaval shunts, PR, and TR. It should be noted, however, that the aortocaval shunt model may lead to PHT over time due to chronic pulmonary overflow, therefore resulting in a combined pressure–volume overload model [13].

A few methods for creating PR in animal models have been described. Kuehne et al. [34], Ersboel et al. [11], and Smith et al. [12] used a stent placed across the pulmonary valve in pigs, thus creating free PR. Moreover, in a study on pigs, Bove et al. used excision of a pulmonary leaflet and infundibulotomy with TAP, as well as a combination of the two methods. All procedures were performed using a right heart bypass [35]. Furthermore, in a study performed on pigs, Agger et al. used external plication sutures placed through the wall of the pulmonary trunk and around the hinge points of the pulmonary valve leaflets, thus preventing coaptation of the leaflets. The procedures were performed without the use of extracorporeal circulation [10]. Similarly, Reddy et al. used external plication sutures to tether two pulmonary leaflets in a murine model, which resulted in a hemodynamically significant PR [3].

Two studies have described RV volume overload caused by PR in sheep. Yerebakan et al. used annulotomy and TAP, followed by transection of the anterior pulmonary leaflet through the patch, without the use of CPB [9,36]. The technique described by the authors differed from ours in that the TAP was initially sutured over the intact RVOT and PA, after which an opening in the patch was created, through which the RVOT was incised and the pulmonary annulus bluntly transected using scissors. Additionally, the authors did not specify how they managed bleeding during the RVOT and PA transection. In contrast, our approach utilized a Satinsky clamp to control bleeding in the operative field, allowing for clear visualization of the anatomy during the RVOT and pulmonary annulus transection. Moreover, our method facilitates augmentation of the RVOT by the entire width of the patch, whereas the technique described by Yerebakan et al. results in a partial loss of patch width due to its overlap with the RVOT and PA wall.

On the other hand, Gray et al. excised two pulmonary leaflets without CPB but with temporary occlusion of the caval veins and PA with snares [37]. In contrast, we used lateral clamping of the PA using a Satinsky clamp to control bleeding during leaflet perforation.

Avoiding extracorporeal circulation is a great advantage, as it eliminates all risks and complications related to CPB. It reduces the duration of the surgery, lowers the risk of bleeding, and is less expensive [10]. All the procedures described in our study were performed off-pump, on a beating heart, and no intra-/postoperative bleeding or neurological or renal complications were registered.

To provide a clear and concise overview of the outcomes of the different surgical procedures, we summarize their key benefits and drawbacks in Table 3. This table highlights the essential findings from our study and serves as a reference for comparing the strengths and limitations of each technique.

A potential limitation of our leaflet perforation technique is that it does not replicate the free PR and RVOT distortion observed following corrective surgery for TOF. Nonetheless, this novel technique can serve as an alternative method for studying the pathophysiological effects and ventricular remodeling associated with RV volume overload. By standardizing the perforation size in the leaflet perforation group and using uniform patch dimensions (2 cm width) in the TAP group, we achieved moderate-to-severe PR in both groups, ensuring highly reproducible results.

### 5.4. Complications

In our study, the procedure-related complications were dominated by arrhythmias, especially in the PAB group (Table 1), during band tightening; most of these were resolved by loosening the band and attempting to gradually and slowly tighten it again. Pharmacological treatment was necessary in only one case of AF.

Sudden cardiac death attributed to arrhythmic events occurs in 2–8% of patients with TOF in the postoperative period. Increased RV systolic and diastolic pressures, among other factors, have been identified as potential risk factors [38].

In a study on pigs, Zelster et al. showed that both PAB and pulmonary valvotomy increased the likelihood of inducible atrial arrhythmias, while pulmonary valvotomy increased the risk of inducible ventricular tachycardia/ventricular fibrillation in the chronic setting. The authors showed that an increased RV end-diastolic pressure was a risk factor for both atrial and ventricular arrhythmias [38].

While determining the exact association between arrhythmias and acute pressure or volume overload of the RV was beyond the scope of our study, it is interesting to note that the PAB group presented more arrhythmic events than the other two groups (Table 1), especially during band tightening. This might further indicate that the RV tolerates acute increases in afterload rather poorly—not only hemodynamically but also electrically—and is worth further research. The bradycardia and ST segment elevation associated with hemodynamic instability that occurred in four out of six PAB animals were likely a sign of acute RV failure, indicating an increase in afterload beyond RV tolerance. Moreover, as postoperative 24 h ECG registrations were not available in our study, the one death that occurred was assumed to have been caused by an arrhythmic event, considering that the animal presented AF intraoperatively and therefore, was susceptible to the development of arrhythmias, as well as considering the nonsignificant pathology exam.

This early postoperative death highlights the inherent challenges of PAB in inducing RV pressure overload. Achieving a balance between achieving sufficient banding to induce RV failure and avoiding acute complications, such as arrhythmic events, remains a critical consideration. Ukita et al. addressed these challenges by employing a staged approach with progressive PAB, allowing for a more controlled increase in RV afterload and reducing the risk of acute decompensation. Despite these precautions, the authors still reported one animal death at week 10 post-surgery, illustrating that even a gradual and controlled approach does not eliminate the risk of adverse outcomes [24].

These findings underscore the complexity of developing experimental models of RV failure, where both acute and late complications remain significant challenges, regardless of the technique employed.

Finally, except for one case of AF that required pharmacological cardioversion and the abovementioned ST segment elevations, the arrhythmic events were benign and self-limiting, without hemodynamic instability.

The other intra- and postoperative complications reported in this study were easily manageable and were cardiac surgery-related rather than PAB- or PR-related, as they were complications that occur frequently in open heart surgery or other prolonged surgeries in ruminants.

The long-term impact on RV hemodynamics and the efficiency of inducing chronic RV failure will be reported in a future study.

Study limitation: One limitation of this study is the relatively small sample size. However, this was adequate to demonstrate the feasibility of the surgical techniques and anesthetic protocol. To prioritize animal welfare, a larger sample was avoided, in alignment with the 3R principles—replacement, reduction, and refinement. Additionally, the study did not include a “sham” procedure group, which could have provided further insights into the procedural impact on hemodynamic and functional changes. This limitation was due to ethical considerations to minimize the number of animals used. Future studies could incorporate a sham procedure to better isolate and analyze the effects of the interventions.

Another limitation is the lack of preload-independent measurements provided by the conductance catheter-derived pressure–volume loops. While fascinating from a theoretical standpoint, providing extensive information regarding cardiac mechanoenergetics, their use in clinical practice is still limited. We aimed to use reliable but reproducible methods that can be easily translated into clinical practice. The Swan–Ganz catheter has been used for over 50 years and remains the cornerstone of invasive hemodynamic monitoring. Pulmonary artery catheterization can be performed at the patient’s bedside, and it offers valuable information regarding heart function that surpasses simple pressure measurements. It offers derived metrics, such as ventricular stroke work, an approximation to that derived from the PV loops, which has been shown to correlate with patient prognosis in clinical practice.

## 6. Conclusions

All three surgical techniques described in our study proved to be successful in achieving pressure and volume overload. Acute pressure overload via PAB led to a significant elevation in RV pressures and a significant decline in SVI and CI, associated with reduced ScvO_2_ and elevated lactate levels, suggesting low cardiac output.

Furthermore, we describe a novel off-pump experimental surgical method for creating PR via pulmonary leaflet perforation using an aortic punch, with comparable results to those for TAP augmentation in achieving acute volume overload. Both procedures resulted in moderate-to-severe PR and an incremental trend in RV pressures, as well as CI and SVI.

The results of our study provide reproducible methods for achieving RV failure through pressure and volume overload in an ovine model. This facilitates an in-depth understanding of RV adaptation in different CHDs, supporting the development of research-based treatment strategies and their translation into clinical practice.

## Figures and Tables

**Figure 1 biology-14-00170-f001:**
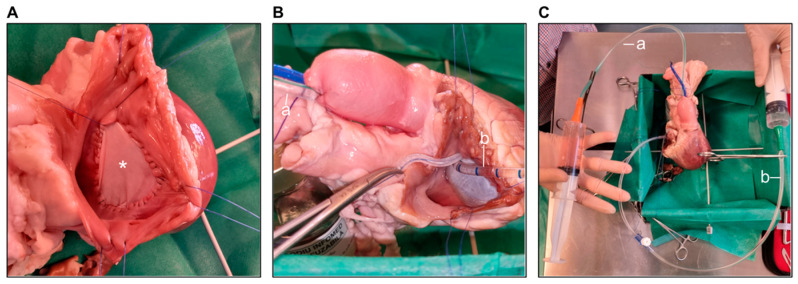
(**A**) The tricuspid valve sealed with a pericardial patch (asterisk). (**B**) The pulmonary artery closed using a purse string suture, through which an aspiration cannula (a) was introduced. Similarly, an aspiration cannula (b) introduced in the RV through an opening between the pericardial patch and the tricuspid annulus, secured by a purse string suture. (**C**) Both aspiration cannulas (a, b) connected to 50 mL syringes.

**Figure 2 biology-14-00170-f002:**
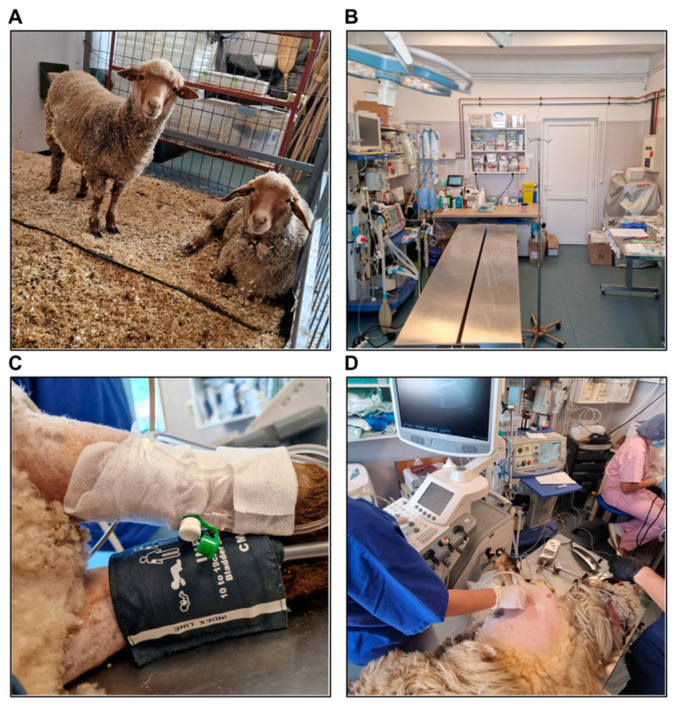
(**A**) The indoor shelter. (**B**) The preoperative preparation of the operating room. (**C**) Peripheral venous access. (**D**) The performance of preoperative transthoracic echocardiography.

**Figure 3 biology-14-00170-f003:**
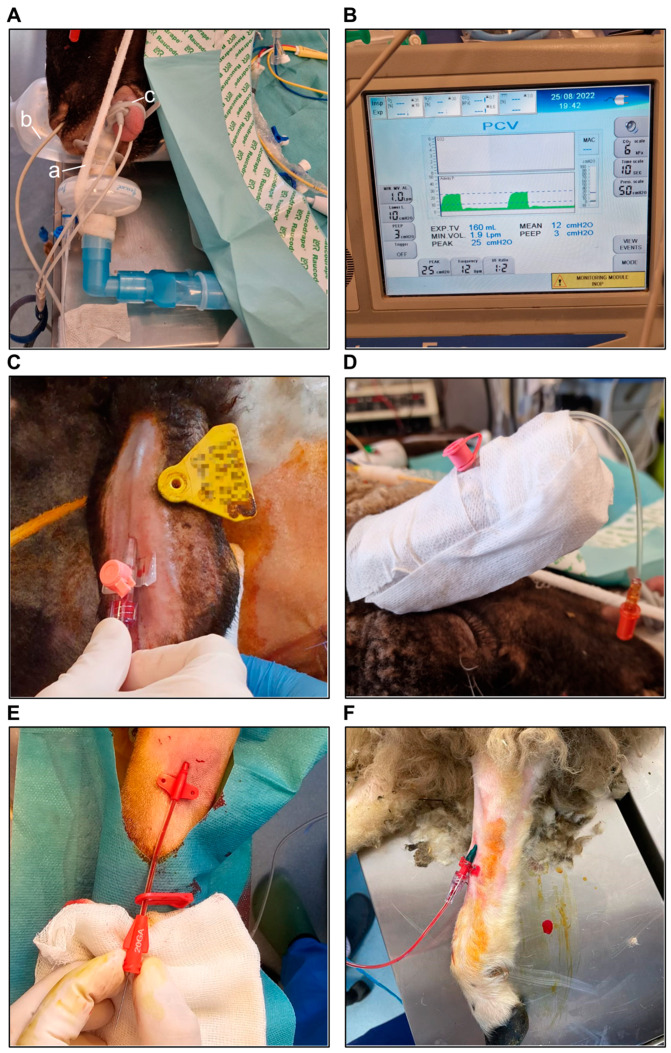
(**A**) The endotracheal tube introduced through the mouthpiece and connected to the ventilator circuit via a bacterial/viral filter (a). The esophageal temperature probe inserted into the esophagus after intubation (b). The pulse oximeter switched to a lingual pulse oximeter after intubation (c). (**B**) Mechanical ventilation using a pressure-controlled ventilation mode. (**C**) A 20 G peripheral venous catheter inserted into the auricular artery. (**D**) The catheter connected to an extension tube, with a valve for postoperative arterial blood gas sampling. (**E**,**F**) A 20 G arterial catheter inserted into the tibial artery for intraoperative invasive BP monitoring and postoperative arterial blood gas sampling.

**Figure 4 biology-14-00170-f004:**
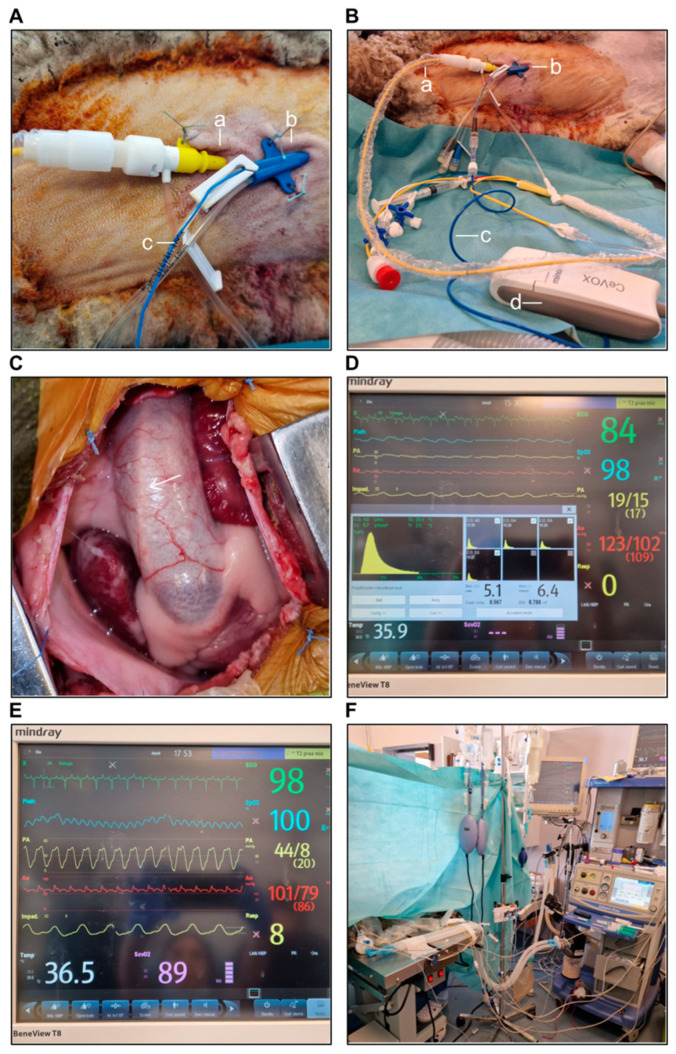
(**A**) The introducer sheath (a) inserted into the left external jugular vein and the central venous catheter (b). The fiber optic probe for continuous ScvO_2_ monitoring inserted into the distal lumen of the central venous catheter (c). (**B**) The Swan–Ganz catheter inserted through the introducer sheath (a). The central venous catheter is shown in (b), along with the fiber optic probe (c) and the CeVOX optical module (d). (**C**) The Swan–Ganz catheter (arrow) advanced into the pulmonary artery. (**D**) The preoperative thermodilution measurement of the cardiac output. (**E**,**F**) Advanced anesthetic monitoring.

**Figure 5 biology-14-00170-f005:**
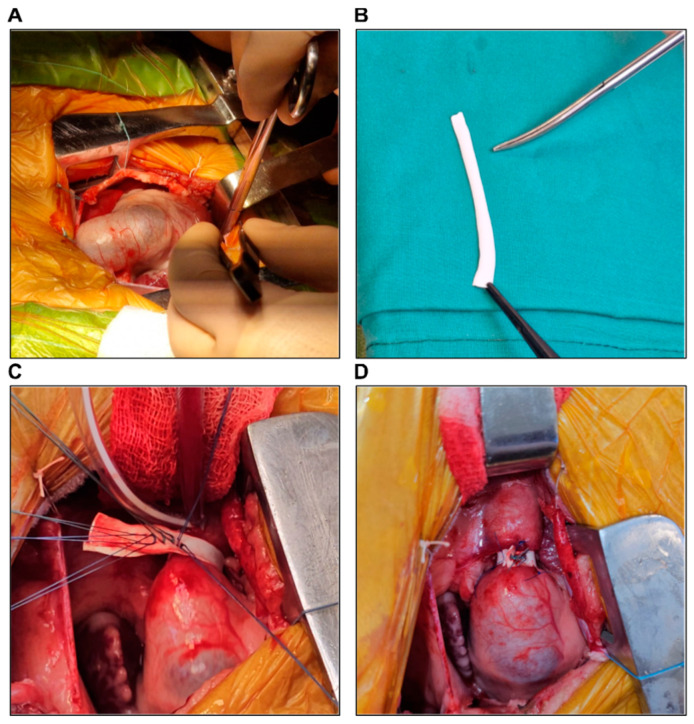
(**A**) Exposure of the pulmonary artery. (**B**) Preparation of the Gore-Tex band. (**C**) Progressive placement of Ti-Cron sutures for pulmonary artery narrowing. (**D**) Final view of pulmonary artery banding.

**Figure 6 biology-14-00170-f006:**
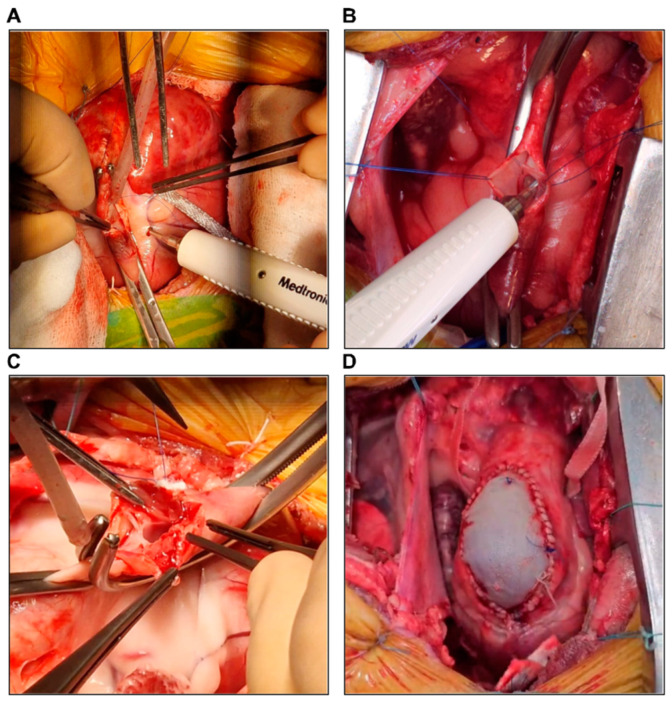
(**A**) Lateral clamping of the pulmonary artery and RVOT and the incision at the level of the pulmonary artery root, exposing a pulmonary leaflet. (**B**) Pulmonary leaflet perforation using a 6 mm aortic punch. (**C**) Lateral clamping and incision of the pulmonary artery, pulmonary annulus, and RVOT. (**D**) Transannular patching using a heterologous pericardial patch.

**Figure 7 biology-14-00170-f007:**
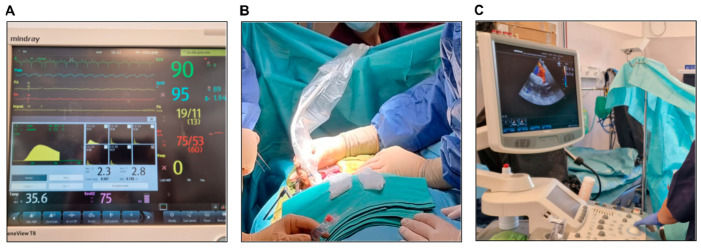
(**A**) Postoperative thermodilution measurement of cardiac output. (**B**,**C**) Postoperative epicardial echocardiography.

**Figure 8 biology-14-00170-f008:**
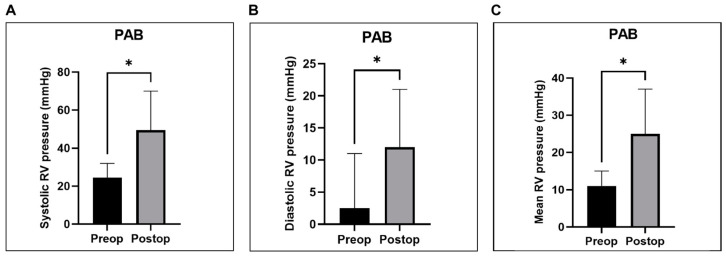
Graphical representation of right ventricular (RV) pressures registered before (pre-op) and immediately after (post-op) PAB. Panels show a significant increase in systolic (**A**), diastolic (**B**), and mean (**C**) RV pressures. * *p* < 0.05. PAB—pulmonary artery banding.

**Figure 9 biology-14-00170-f009:**
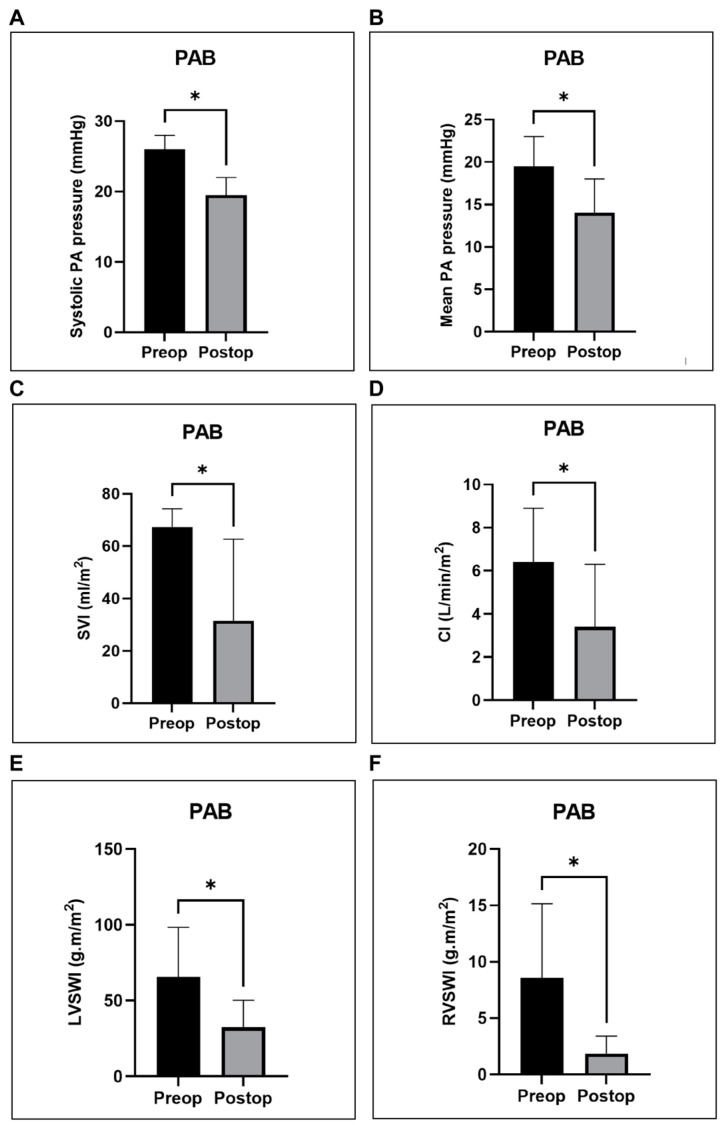
Graphical representation of invasive hemodynamic measurements registered before (pre-op) and immediately after (post-op) PAB using Swan–Ganz catheterization. (**A**,**B**) Significant decrease in systolic (**A**) and mean (**B**) pulmonary artery (PA) pressures. (**C**,**D**) Significant decline in SVI (**C**) and CI (**D**). (**E**,**F**) Significant reduction in LVSWI (**E**) and RVSWI (**F**). * *p* < 0.05. PAB—pulmonary artery banding; SVI—stroke volume index; CI—cardiac index; LVSWI—left ventricular stroke work index; RVSWI—right ventricular stroke work index.

**Figure 10 biology-14-00170-f010:**
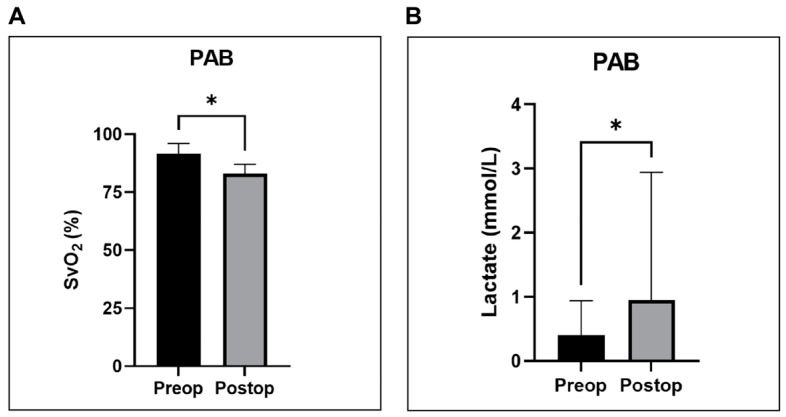
Graphical representation of mixed venous oxygen saturation (ScvO_2_) and serum lactate levels determined before (pre-op) and immediately after (post-op) PAB. (**A**) Significant decline in ScvO_2_. (**B**) Significant elevation in serum lactate levels. * *p* < 0.05. PAB—pulmonary artery banding.

**Table 1 biology-14-00170-t001:** Intraoperative complications.

Complication	Surgical Procedure Type		
	Annulotomy + TAP (*n* = 4)	Pulmonary Leaflet Perforation (*n* = 4)	PAB (*n* = 6)
**Total arrhythmic events**	9	9	22
**Pre-induction (number of cases)**			
** • Sinus bradycardia**	4	2	4
** • SV extrasystoles**	1	2	3
** • AVB grade II type I**	0	1	0
**Pre-procedure (number of cases)**			
** • Ventricular extrasystoles**	1	1	1
** • SVT**	0	0	1
**Intraprocedural (number of cases)**			
** • Bradycardia**	0	0	4
** • SVT**	0	0	1
** • Ventricular extrasystoles**	2	3	3
** • ST segment elevation**	0	0	4
** • Atrial fibrillation**	1	0	1
**Hypothermia (number of cases)**	4	3	5
**Mean central body temperature (°C)**	36.3	36	35.8
**Hyperthermia (number of cases)**	0	1	0
**Tympanism (number of cases)**	3	1	2

SV—supraventricular; AVB—atrioventricular block; SVT—supraventricular tachycardia.

**Table 2 biology-14-00170-t002:** Early postoperative complications (number of cases).

Complication	Surgical Procedure Type		
	Annulotomy + TAP (*n* = 4)	Pulmonary Leaflet Perforation (*n* = 4)	PAB (*n* = 6)
Death	0	0	1
Pericardial effusion	1	2	2
Cardiac tamponade	1	0	0
Pneumothorax	0	1	0
Resuscitated CA	1	0	0
Mild fever	1	1	4
Metabolic alkalosis	2	3	2
Mixed alkalosis	2	1	3
Hypokalemia	3	4	3
Tympanism	0	2	0
Constipation	0	2	2

CA—cardiac arrest.

**Table 3 biology-14-00170-t003:** Key benefits and drawbacks of surgical procedures.

Surgical Procedure Type	Key Benefits	Key Drawbacks
**Pulmonary Artery Banding (PAB)**	- Simple surgical technique - Does not require cardiopulmonary bypass (CPB) - Reproducible results for pressure overload models	- Risk of acute RV failure and arrhythmias during band tightening - Requires precise adjustment to balance between sufficient banding and hemodynamic tolerance
**Pulmonary Leaflet Perforation**	- Novel method with consistent results in inducing moderate-to-severe pulmonary regurgitation (PR) - Less invasive than TAP - Off-pump technique minimizes risks and reduces costs	- Does not replicate free PR and RVOT distortion observed in TOF repair - Limited by perforation size (aortic punch size)
**Pulmonary Annulotomy + Transannular Patching (TAP)**	- Effective in creating significant volume overload - Achieves moderate-to-severe PR - Highly reproducible outcomes - Off-pump technique minimizes risks and reduces costs	- More invasive procedure requiring RVOT incision - Longer chest drainage period and increased fluid accumulation - Higher risk of complications due to procedural complexity

PAB—pulmonary artery banding; RV—right ventricle; CPB—cardiopulmonary bypass; PR—pulmonary regurgitation; RVOT—right ventricular outflow tract; TAP—transannular patching; TOF—tetralogy of Fallot.

## Data Availability

The datasets used and analyzed during the current study are available from the corresponding author upon reasonable request.

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
