# Peer review of "Early Outcomes of Right Ventricular Pressure and Volume Overload in an Ovine Model"

_biology, 2025, doi:10.3390/biology14020170_

Round 1

Reviewer 1 Report

Comments and Suggestions for Authors

Title: Ovine Model of Right Ventricular Failure through Pressure and Volume Overload: Part 1—Surgical and Anesthetic Protocol

Major Comments

-While I understand that the paper focuses on surgical and anesthetic protocols, this paper does not report on any RV functional metric. It may be the author's intent to have a subsequent manuscript report on the results, but I don't think that is the right approach. Without any outcome metrics, it is less valuable to report just on the surgical/anesthetic methods.

-Because of the point above, the Discussion section reads more like a literature review. Most of it is summary of prior works and it's not until the very end where the authors describe their own work. If the authors have more data to present, then they will have more to discuss in the context of these other works

-There's too much discussion on anesthesia, and not as much on the surgical approach. A lot of the anesthetic monitoring steps can be cut down as these are standard and not specific to the surgeries.

-Line 599: while ovine models are scarce, Matthew Bacchetta from Vanderbilt has published extensively on their sheep model of progressive pulmonary artery banding. This work should be compared to the present work and the existing literature

Minor Comments
- PVR is not a good acronym for pulmonic valve replacement (pulmonary vascular resistance)
-Line 622: adjustable PAB is mentioned, but it doesn't cite original research article (ref 13 is a review paper)

Reviewer 2 Report

Comments and Suggestions for Authors

In this paper the authors wanted to investigate right heart failure using an ovine animal model.

The description of the method is exhaustive. The professionalism with which they took care of the animals and induced and maintained anesthesia is admirable. It should be mentioned that the sheep animal model is not a new one. There have been studies on sheep that try to investigate heart failure including right heart failure. Intraoperative motorization is adequate for the study.

But the article is not a research article, it just describes the method. For this purpose there are dedicated journals (ex. Nature Methods).

Only some data related to the occurrence of complications are presented.

The discussion section addresses interesting topics, but they are not focused on the presented results.

I believe that using this method the authors can obtain important results that deserve to be presented in this Journal and I recommend the resubmission of the article when they will have all the results analyzed and discussed.

For this stage I consider that the article is not suitable for publication.

Reviewer 3 Report

Comments and Suggestions for Authors

This is an interesting and well conducted stUdy in sheep describing the anaesthetic and surgical protocols .The sheep were subjected to three different procedures to study methiods of combatiing right ventricular failure;

A considerable amount of information is comprehensively detailed in the text which makes a number of the figures and tables superfluous  e.g. Figs 3 and 7

The paper is far too long and includes too much detail of routine activities Comments.

Lines -52 ,65 and 157 How old were the sheep ? -" Juvenille" can cover a wide variety of ages

Line 191  A lot of the material included here is NOT result .Description of anaesthetic and surgical techniques are Methods.

Line 227 and 248 Atropine is  not esential in any  anaesthetic technique .The justification quoted is not acceptable as in  most species administered after alpha -2 agonists it produces arrhythmias

Lines 262 and 373 The animal was not ventilated   but its lungs were.

Lines 346  It should have been considred essential to monitor Neuro-muscularblockade .Was this performed ?Was the block reversed?

Line 354 Anaesthesia is not reversed -Animals recover once administration has ceased

Line362  Sheep do NOT vomit and regurgitation is part of the digestive process of rumination

Reviewer 4 Report

Comments and Suggestions for Authors

Dear colleagues,

the version presented to the current reviewer had already been reviewed by 2 reviewers; the comments were addressed by the authors, subsequently, the authors have made significant improvements.

What I would like to suggest to make final amendments and make the article more concise:

- the authors are kindly asked to put the "complications" section before the discussion as this seems more logical;

- the authors are also kindly asked to give information regarding the key benefits and drawbacks of each model in a separate table; this will summarize the information given in the "results" and "discussion" and bring reader's attention to the core of the article;

- this experimental work also did not include a "sham" procedure on the animals, which should be addressed in the text and explained in the "limitations" section.

Overall, the manuscript gives a good impression. Substantial work on the article has already been performed after the first revision. The manuscript is recommended for publication after minor revision.

Round 2

Reviewer 1 Report

Comments and Suggestions for Authors

MDPI  biology-3316678

The authors have returned with data to describe the hemodynamic indices after the surgery. However, the efficacy of their surgical approach is questionable, as there are no discernible changes to the RV for two of the surgical approaches, and some of the data for PAB do not quite make physiological sense. I suggest that the authors have a way of quantifying success of their surgical approach to confirm that pressure/volume overload is achieved.

Major Comments

Figures 8 and 9: I find it difficult to understand why the RV pressure goes up but the PA pressure goes down after PA Banding. The two changes should directly correlate. The systolic pressure of RV and PA should be essentially the same- which seems to be the case at pre-op but not at postop.

There are no discernible changes in RV functional metrics for pulmonary leaflet perforation and pulmonary annulotomy. How would you define a successful surgical experiment?

If there are no observable changes immediately postop, there should be a follow-up assessment to see if the changes are more noticeable days/weeks out from the surgery.

The discussion is too lengthy. There's little discussion of the authors' own work and too much about the literature. I find that this would improve, if authors have more data to present on the efficacy of their surgical approaches

Minor Comments
Line 442 in the revised/tracked version: provide reference for Trusler's formula

Line 496-497: 10-minute soaking in glutaraldehyde doesn't seem like sufficient time for crosslinking. do authors have any justification for this duration?

Figures 8 and 9: I find it difficult to understand why the RV pressure goes up but the PA pressure goes down after PA Banding. The two changes should directly correlate. The systolic pressure of RV and PA should be essentially the same- which seems to be the case preop but not at postop.
Also, why do you only show PAB results as a graph, and the rest as part of a supplementary table?

I think some of the postoperative outcome results (Table S8 and S9) can be part of the main text since you discuss in length about it i the main text
